# Wild Boars’ Selective Capture with IoT and Electronic Devices Solutions for Innovative, Sustainable and Ethical Management

**DOI:** 10.3390/s25072071

**Published:** 2025-03-26

**Authors:** Maria Teresa Verde, Luigi Esposito, Francesco Bonavolontà, Oscar Tamburis, Annalisa Liccardo, Nadia Piscopo

**Affiliations:** 1Department of Electrical Engineering and Information Technology, University of Napoli Federico II, 80137 Napoli, Italy; mariateresa.verde@unina.it (M.T.V.); francesco.bonavolonta@unina.it (F.B.); aliccard@unina.it (A.L.); 2Department of Veterinary Medicine and Animal Productions, University of Napoli Federico II, 80137 Napoli, Italy; nadia.piscopo@unina.it; 3Institute of Biostructures and Bioimaging National Research Council Naples, 80145 Naples, Italy; oscar.tamburis@cnr.it

**Keywords:** wild boar management, population control, IoT-based selective capture trapping, animal welfare, sustainable ecosystem management

## Abstract

The growing population of wild boars (*Sus scrofa*) in all highly anthropized countries represents a current challenge for the protection of ecosystems, agriculture and urban environments. This study introduces an innovative capture solution based on IoT systems designed to enable the selective capture of sub-adult wild boars in an ethical manner and minimize stress and bycatch. Conducted over five years in a Natura 2000 area in Campania, Italy, the research integrates advanced technologies, including AI-based infrared cameras, LoRa communication and autonomous feeding systems, to monitor, control and operate a specially designed selective cage trap. The results obtained demonstrate how technological innovation improves wildlife and hunting management by selecting younger animals without interfering with group dynamics. Selective capture ensures healthy population control, does not conflict with hunting and reduces pressure on habitats, especially if these fall within areas of particular importance for European biodiversity.

## 1. Introduction

The conflict between wild animals and humans has recently focused on some species such as large carnivores in Africa and America [1], the wolf (*Canis lupus*), the bear (*Ursus arctos*) and the lynx (*Lynx lynx*) in Europe [2,3]. Their negative impact is related to predation on domestic livestock [4]. Besides these, the wild boar is particularly known due to its association with damage to agriculture [4,5], road accidents [6,7] and the invasion of cities [8]. The problem related to wild boars is currently associated with their population explosion [9,10], since they are considered one of the most invasive species on the planet after humans [11,12]. The species *Sus scrofa* present in the Palearctic Region has also spread over time in the Italian peninsula. After the Second World War, its distribution underwent a significant contraction [13] and, starting from the 1950s, there was a large introduction of animals from north-eastern Europe [13,14,15]. Currently, due to uncontrolled hybridization with domestic pigs [16,17], the animals present on Italian territory have all undergone a marked influence on their phenotypic expression [18], as well as on their ability to occupy any type of habitat [13]. Their expansive capacity became evident when they started to graze outside the tree vegetation, and on cultivated lands and at waste collection points in urban areas. The urgency, therefore, arises to find out ways to achieve (i) adequate monitoring of wild boars’ movements, and (ii) allow selective, safe and remotely controlled capture—even in areas without network coverage or Internet access.

The present work is based on the results of the European Rural Development Policy 2014–2020 project (PSR 2014–2020 “S.U.S. Campania” CUP B58H19004460009) whose main objective was to address the issues related to the possible profitability deriving from natural resources when these overlap with agricultural and livestock activities, producing ecosystem services or disservices [19]. The entire innovative process developed in the project was based on the design and implementation of a remote monitoring system featuring the use of IoT–based sensors [20], to allow optimal positioning of an ad hoc–designed trap cage. Remote monitoring of trap cages via video surveillance or motion sensors, unlike the use of drones [21], allows operators to immediately know when a trap is activated, minimizing animal stress derived from capture [22]. Current trap cage technologies involve non-selective capture and do not consider the importance of age or size. Traditional traps have several drawbacks such as the lack of selectivity, the risk of involving other wild animals [23] and the impossibility of ensuring the safety and welfare of captured animals [24,25,26].

The main goal was to collect real-time data on the behavior and movements of wild boars to provide precise indications before, during and after the capture of animals, such conditions influence the quality of the innovative supply chain and ensure best practices for animal welfare.

## 2. Materials and Methods

### 2.1. Experimental Design and Project Phases

The experimental period lasted 60 months (2019–2023) with the main objective of addressing the problems related to the presence of wild boars in a specific area of the province of Avellino (Campania, Italy). The experimental scheme proposes an innovative process for the management of the wild boar population. All the methods proposed to date are based on the indiscriminate numerical reduction (no distinction between age classes and sex) carried out with firearms or by capturing the greatest possible number of subjects.

Our work proposes a demographic reduction of the population by operating only on the young classes, so as not to disturb the family groups. In previous works [26,27], the presence of sub-adult wild boars (6–12 months of age) in the sample area was described by camera trapping [27].

The observational study proved useful to focus attention on the passage area and the age classes of the animals at a fixed point. The camera traps were positioned as described in the following way to capture images of the wild boars as they crossed the observation point at either the entrance or the exit. Preliminary observations allowed us to identify the precise point for the construction of a selective capture cage. Once the positioning of the cage was established, various electronic IoT-based devices were integrated to ensure efficient and safe operation.

The action scheme is summarized in the next sections.

In a preliminary phase [27], the passage of wild boars in the area used for the construction of the selective capture cage was demonstrated. The farm, leader of the PSR 2014–2020 S.U.S. Campania project, has an extension of about 10 hectares (40°50′40.96″ N; 14°37′36.69″ E) and is included in the SCI/SAC IT8040013 “Monti di Lauro” whose task is to protect particular forms of plant and animal biodiversity.

The selection cage was designed to allow good management of living beings in the Natura 2000 area, including human activities and the presence of invasive animals such as wild boars whose environmental impact is described only as negative.

The activity of the cage was, therefore, monitored through external camera traps and the integrated IoT system.

Specialized operators recorded and archived all the data obtained during the experimental period.

Figure 1 describes how the three phases of the field study were articulated.

In the first phase, the main objective was to ascertain the presence of wild boars in the study area. Subsequently, the number of subjects and the structure of the population on which it was intended to intervene were determined [28]. The use of 19 camera traps made it possible to produce digital maps (Figure 2) related to the distribution of wild boars in the point area (4.70 hectares) and in the surrounding agroecosystem (61 hectares). The camera trap model used is a “Spromise”^®^ Full HD + SMS (Digital Trail Camera S358). The weight of each camera trap is 0.38 kg, while the dimensions are 15.3 × 12.4 × 9.0 cm. The detection range and the flash range of camera traps are 25 m and 18 m, respectively. The functions were set to perform high-resolution images (12 MP), infrared night lighting and high-mode motion sensors (PIR). During the overall experimental period, 405 individual assessments were performed on the animals captured by the camera trap videos, 302 shots were examined and 3619 min of recorded images were studied [27].

In the second phase, an IoT-based video surveillance system was applied to the cage designed to capture and retain the wild boars.

The system’s features include:**A smart infrared thermal camera** capable of detecting temperature variations and monitoring defined perimeters, with real-time analysis of anomalies and movements. This system, installed on one of the poles on the perimeter of the capture cage, allowed the detection of the type of animals standing inside the cage. The counting of wild boars and the definition of their age classes was carried out using the GitHub algorithm—Ultralytics YOLO8 trained to recognize the morphological characteristics of wild boars (detection, object tracking, image classification). In the case of wild boars belonging to the right age class (sub-adults), the electronic processing unit allowed operators to activate the closing of the entrance doors to, subsequently, access the cage with the appropriate equipment necessary for the correct containment of the animals of interest. Conversely, in the case of the presence of wild boars of a different age class and/or of animals of other species, no action was performed [22,23,24,25,26,27].**An electromagnetic lock is installed on each door to keep it normally open.** The system consists of an electromagnet mounted on the door frame and a metal plate connected to the door via a cord. When powered, the electromagnet generates a strong magnetic field that holds the metal plate in place, keeping the cord taut and the door open. In the event of a power outage, the magnetic field dissipates, releasing the metal plate and the cord simultaneously. This mechanism ensures the door closes quickly and securely.

An electronic logic unit is responsible for controlling the opening and closing of the doors. It consists of a board based on the STM32L072CZ microcontroller and the LoRa (Long Range) SX1272 module, flanked by a relay connected to a GPIO (General Purpose Input/Output) of the board.

Under normal operating conditions, the board keeps the relay in the normally closed (NC) state, providing a constant 12 V power supply to the electromagnetic locks. These locks generate a magnetic field that holds a metal plate, dissipating a current of 0.158 A (Adc) and allowing the doors to be kept open.

When the SX1272 LoRa module receives a remote command, the STM32L072CZ microcontroller sends a signal to the GPIO to switch the relay to the open (NO) state. This stops the current flow to the electromagnetic locks, resetting the magnetic field and releasing the metal plate. Releasing the plate causes the doors to close quickly and safely, as mentioned above.

The logic unit, thanks to the use of the STM32L072CZ, guarantees low energy consumption, as well as the ability to control the cage closing mechanism from long distances via communication with the LoRa protocol. This allows for remote operation, reducing disturbance to the animals [19].

**Remote control of door locking thanks to LoRa Ping-Pong communication and end-to-end encryption (E2EE).** The communication system consists of: (1) a remote device (telecommand), i.e., a portable LoRa controller, with a button to send commands, and (2) the electronic logic unit that controls the electromagnetic locks. Communication between the remote control and the logic unit uses AES-128 encryption to ensure security (E2EE). Each message sent is encrypted and includes a unique sequence number and timestamp to prevent replay attacks. When the user wants to close the doors, he presses the button on the LoRa remote control to start the procedure. The remote control then sends an encrypted Ping message containing the “ALERT” command, a sequence number and a MAC (Message Authentication Code) to ensure integrity (payload: {“command”: “ALERT”, “sequence”: 12345, “MAC”: “checksum”}). The logic unit receives the message, decrypts it using the shared AES key and verifies the sequence number and the MAC. If everything is valid, the STM32L072CZ microcontroller switches to the “ALERT” state and sends a Pong as a response, confirming the reception of the initial command ({“response”: “PONG”, “sequence”: 12345, “MAC”: “checksum”}). On the LoRa remote control, the lighting of a Yellow LED confirms to the user the correct reception of the Pong and the “ALERT” state of the logic unit, which is ready to receive the confirmation of closing the doors. When the button is pressed, the remote control sends a second encrypted message with the final command “RELAY_ON”, a new sequence number and the MAC (payload: {“command”: “RELEASE_ON”, “sequence”: 12346, “MAC”: “checksum”}). The logical unit receives the message, decrypts it, verifies the sequence number and the MAC and, if everything is valid, executes the command. The STM32L072CZ microcontroller sends a signal to the GPIO to switch the relay to the open (NO) state. This cuts off the power to the electromagnetic locks, bringing the magnetic field to zero. The metal plates are released, causing the doors to close quickly and safely. In practice, the system offers numerous advantages, including a high level of security guaranteed by AES-128 encryption, which prevents unauthorized access and ensures the integrity of the transmitted data. The double confirmation required to execute the command minimizes the risk of human errors or accidental activations, increasing the reliability of the system. Furthermore, LoRa communication enables long-range remote control with low power consumption, making the system highly efficient. The immediate feedback provided to the user ensures that the command has been correctly received and understood by the logic unit. These features make the system ideal for critical applications, such as managing cage doors, where closure at an inopportune moment could compromise the entire operation.

Finally, to ensure maximum robustness in communication, the LoRa communication parameters have been set as follows: Spreading Factor (SF12) and Bandwidth (BW250).

Using a **wireless Wi-Fi bridge**, the infrared camera transmits real-time images of the inside of the cage to a remote station, making it an ideal solution for monitoring areas without telephone or Internet coverage and allowing specialized operators to intervene promptly when necessary [20]. In this specific case, the effective distance was 563 m in a straight line, with obstacles such as tall trees that could attenuate the signal (Figure 2). To ensure the connection, only two directional antennas were used, one for transmission and one for reception, without any intermediate repeaters. Figure 2 also indicates the locations of the devices. Point 1 marks the position of the trap, while Point 2 corresponds to the remote receiving station. Despite environmental challenges, the system maintained a stable and effective transmission.**Autonomous power supply:** the cage was equipped with an autonomous power supply system, which makes it possible to operate in remote areas without access to electricity networks. In particular, an isolated system was created with a solar system sized to power the three 12 V electromagnetic locks 24 h a day, the infrared camera with one side of the Wi-Fi bridge, and the door management logic unit. The overall consumption of the system, estimated at around 1.5 A (equal to 36 Ah per day), required the installation of a 72 Ah lead-acid battery (with a Depth of Discharge of 50%). To ensure efficient charging, a solar panel with a nominal power of 120 W was chosen, capable of supplying a current of around 7.2 A in the 5 h of useful sunshine per day, taking into account climatic variations. The system was completed with a 10 A charge regulator, compatible with the 12 V circuit, to manage current peaks and optimize battery charging. This sizing has ensured a stable and continuous operation of the system, even under conditions of intense use and environmental variability.

The third phase was focused on the assessment of the meat quality from captured animals, to underscore the relative Strengths, Weaknesses, Opportunities and Threats (SWOT) of the entire process, and highlight accordingly the potential in terms of innovation as well as critical issues [28].

### 2.2. Selective Trap Cage System Features

To obtain useful management data, we used measuring instruments equipped with sensors capable of returning the measured value in a standardized and repeatable way. All measurements are immediately recorded in a dataset on a portable PC [29,30].

Schematic view of the selective trap cage:

Figure 3 depicts a comprehensive view of the innovative trap cage, designed for the selective capture of sub-adult wild boars. The cage’s frame was constructed with reinforced materials, ensuring the durability and strength necessary to withstand the physical power of wild boars, which are known for their strength and determination. The overall design is modular, allowing for easy transport, assembly and maintenance in challenging environments such as forests, mountainous regions or rural areas.

This cage is equipped with a variety of technological enhancements aimed at improving efficiency and selectivity. The layout of the cage provides sufficient space for the wild boar of a specific dimension (sub-adult) to enter and move freely before the selective capture mechanism is activated. This space allows for stress-free entry, reducing the likelihood of panic or injury to the animal. Further, the structure ensures animal welfare, as it provides a safe environment capable of preventing injury during capture.

Detail of the selective overhead door:

Figure 4 shows the complete external structure of the capture cage (a) positioned in the field ready for operation. Detail (b) focuses on the overhead door that plays a key role in the selective capture of wild boars. Unlike traditional cage doors which normally close like a guillotine and allow entry to all age classes, this selective door is specifically designed to allow entry only when sub-adult wild boars are present in the cage.

The doors are located precisely along the operational perimeter of the Smart infrared camera, which accurately detects the size of the animals passing through. Thanks to the high image quality, both in daytime and nighttime conditions, the system clearly identifies the species of the animal entering the enclosure and determines whether it is a sub-adult or an adult individual.

These criteria help ensure that adult boars or other unintended species are not captured, aligning with both animal welfare concerns and the objectives of targeted population management. The overhead door operates with a smooth, mechanical action that minimizes noise and stress for the animal. When operators, using the infrared camera, detect the presence of a significant number of sub-adult wild boars inside the enclosure and determine that it is the ideal moment to proceed with their capture, they press the button on the LoRa remote control. The first press “alerts” the logic unit, and the second sends the actual closure command. Upon receiving this second signal, the logic unit, through the activation of the control relay, interrupts the circuit powering the electromagnetic locks and triggers the closure of the doors. The closure is sudden, simultaneous and rapid, preventing the wild boars from understanding what is happening in time and leaving them no chance to escape. This selective approach not only protects non-target species but also avoids unnecessary captures that could compromise the well-being of larger adult boars, who may become more aggressive or injure themselves while attempting to escape.

Infrared camera with intelligent features:

Figure 5a,b shows cases of the integration of the smart infrared camera system equipped with advanced features. This technology is one of the central innovations of the trap cage, allowing for precise and intelligent monitoring of the animals that enter the cage. The infrared capabilities enable the system to function effectively even in low-light conditions, such as at night or in densely forested areas where wild boars are most active.

Improving the integration between the level of intelligence and the trapping facilities is essential to ensure that the trap does not mistakenly capture non-target animals, such as small wild or domestic animals, thus reducing by-catch and aligning with ethical wildlife management practices [22].

Additionally, the video recordings captured by the camera can be used to estimate the population size by counting the number of wild boars observed within the monitored area. This data is invaluable for population management and research purposes, offering detailed insights into the movements, behavior and demographics of wild boar populations in the region. By analyzing the footage, researchers can also assess patterns such as group dynamics, habitat preferences and seasonal variations in activity.

IoT electronic unit for overhead door closure:

In Figure 6a,b, the detailed internet of things (IoT) electronic unit is displayed, which is responsible for controlling the overhead door mechanism remotely.

The LoRa (Long Range) communication technology used by the electronic unit operates effectively in remote areas with little to no cellular coverage. This feature is particularly beneficial in mountainous or forested regions where wildlife management is often needed but conventional communication networks are unreliable [31,32,33,34]. LoRa allows the trap to be monitored and controlled without requiring human presence on-site, thus reducing the need for physical interventions that could disturb the animals or put operators at risk.

Electronics are also designed to be energy efficient, drawing minimal power to maintain long-term operation in areas with limited access to power sources [20].

Figure 7a,b and Figure 8a,b illustrate the wireless communication bridge system.

This system deploys Wi-Fi allowing for real-time monitoring of the cage’s interior. Operators can view live footage from the infrared camera, receive updates on the trap’s status and manually activate the door closing remotely via the LoRa communication channel, minimizing the intervention time of the operators and, therefore, the time spent in the selective capture cage, reducing stress and improving the minimum level of animal welfare.

Additionally, the communication bridge is designed to be compatible with mobile devices, meaning operators can monitor and control the trap from anywhere using a smartphone or tablet. This feature allows for rapid intervention if needed, ensuring the timely and humane handling of captured animals [21].

### 2.3. Statistical Analysis

All the collected data were archived and subsequently analyzed to verify the expected results. The images returned by the camera trapping were classified according to the age classes and the size that prevented or allowed entry to the selective capture cage. Through the IoT system, the images obtained were further selected by showing the details of the animals that passed through the video-monitored area.

The data sorting allowed us to identify four age classes according to different authors [35], and the frequencies obtained, on the total number of observations per month, were compared with the non-parametric *Chi-square* test.

## 3. Results

In the overall experimental period, which began on 1 January 2019 and ended on 31 December 2023, it was possible to carry out 582 individual assessments on animals captured via video from camera traps, while the images obtained through the IoT system allowed us to further select the images, producing the details of 461 animals that passed through the video-monitored area.

The system trained to recognize the morphological characteristics of wild boars and the experience of the authors showed the sensitivity was 74.57% while the specificity was lower (57.22%).

Table 1 summarizes the number of wild boar subjects captured from 2019 to 2023 by camera traps (CT) and after to be analyzed with the IoT system.

The data obtained using the IoT system are utilized to categorize animals by age group and includes the total number of subjects recorded for each year (Table 2).

The table illustrates the fluctuating numbers of wild boar subjects captured by camera traps across different age groups over a five-year period. The total number of captured subjects increased in 2021 with a count of 109, followed by a decline to 73 in 2022 and a subsequent peak in 2023 with a count of 118 animals. Notably, from August 2022 through December 2023, only sub-adult wild boars were captured within our prototype trap cage.

Figure 9 shows that if the sub-adults (G2 and G3), the only ones that can enter the selective capture cage, are removed from the existing wild boar population, the remaining population is contained at 40%, 56%, 51%, 38% and 46%, respectively, in the years 2019, 2020, 2021, 2022 and 2023 (*p* < 0.05).

This targeted trapping aligns with our focus on selectively managing the wild boar population by capturing younger animals, ensuring more controlled and humane wildlife management practices.

## 4. Discussion

The “Wild Boar Capture System and Method”, offers several key advantages over existing technologies in terms of wild boar management, starting with its selective capture capability. By leveraging advanced IoT technologies, the trap cage can specifically target sub-adult wild boars while minimizing the capture of adults and non-target species. This targeted approach not only enhances the effectiveness of population control efforts but also aligns with animal welfare principles, ensuring that the management practices are both ethical and efficient.

Secondly, the remote monitoring features greatly enhance operational safety and reduce the risk of accidents for operators and others who might inadvertently come into contact with the cage trap. Overall, the technological advances achieved through the invention ensure continuous monitoring, timely intervention, animal welfare actions and optimal wildlife management and propose an effective and adaptable process solution for the control of wild boar populations.

Lastly, the cage’s applicability with autonomous power supply and advanced communication technologies makes it particularly effective in rural and mountainous regions where traditional infrastructure is sparse or absent.

Thanks to the integration of advanced technologies such as IoT sensors, microcontroller technology and artificial intelligence (AI), the trap system allows for remote monitoring and management of wild boars [36]. Capture and monitoring operations are conducted through a LoRa Ping-Pong network, which enables wireless communication over long distances without the need for traditional network infrastructure. This means that operators can manage and control the traps without being physically present at the site, reducing the risk of accidents and direct contact with the animals. By integrating infrared cameras, the trapping system provides continuous and secure supervision of wild boar population control operations, demonstrating how technological innovation can contribute to the safer and more efficient management of wild species. This technology of remote monitoring not only optimizes the capture process but also enhances operator safety, as they can intervene promptly if necessary thanks to continuous visibility of the trap contents.

## 5. Conclusions

The results obtained demonstrate that technological innovation helps to identify the different age classes of wild boars and that the quantification of younger animals (G1), compared to adult animals (G4), is the crucial point for correct wildlife and hunting management. Being able to select captures by preferring younger animals (G1) without interfering with group dynamics ensures healthy demographic control as these, normally, would not be used by hunters. Subsequently, the animals could be inserted into a controlled supply chain, reducing the pressure on habitats, especially if these fall within areas of particular importance for European biodiversity. After capture, the animals could be sent for slaughter under the control of official veterinary bodies, guaranteeing both food safety and hygiene-health control as well as demographic control.

The invention described marks a significant leap forward in the management of wild boar populations, combining cutting-edge technology with a commitment to ethical and sustainable practices, offering a solid solution to the agricultural and environmental challenges posed by wild boars. This innovative cage trap not only mitigates the damage caused by these animals but also introduces new ways to manage wildlife. Its advanced features ensure that the management process is safe and efficient, while also prioritizing animal welfare. The entire innovative process, based on the use of IoT sensors and identified as “short supply chain S.U.S. Campania”, has allowed us to propose an alternative to the current methods of the containment of the wild boar populations (sport and selection hunting) but also to attribute an economic and qualitative value to the meat [15,37].

## 6. Patents

This work was based on a “System and method for capturing wild boars” proposed in the PSR 2014/2020 Campania project. Intervention type 16.1.1 “support for the establishment and operation of European Innovation Partnership for Agricultural Productivity and Sustainability (EIP-AGRI) and Operational Groups (OGs)”. Action 2 “Support for Operational Innovation Projects”. “Technological use and new innovative practices for the management, control and economic valorization of wild boars (Sus scrofa) in a sustainable manner in the Campania Region”. S.U.S Campania (CUP B58H19004460009). The capture cage was submitted as an invention to the Italian patent (Italy—Invention Patent No. 102023000019425 21 September 2023 “System and method for capturing wild boars”, University of Naples Federico II—Department of Veterinary Medicine and Animal Production).

## Figures and Tables

**Figure 1 sensors-25-02071-f001:**
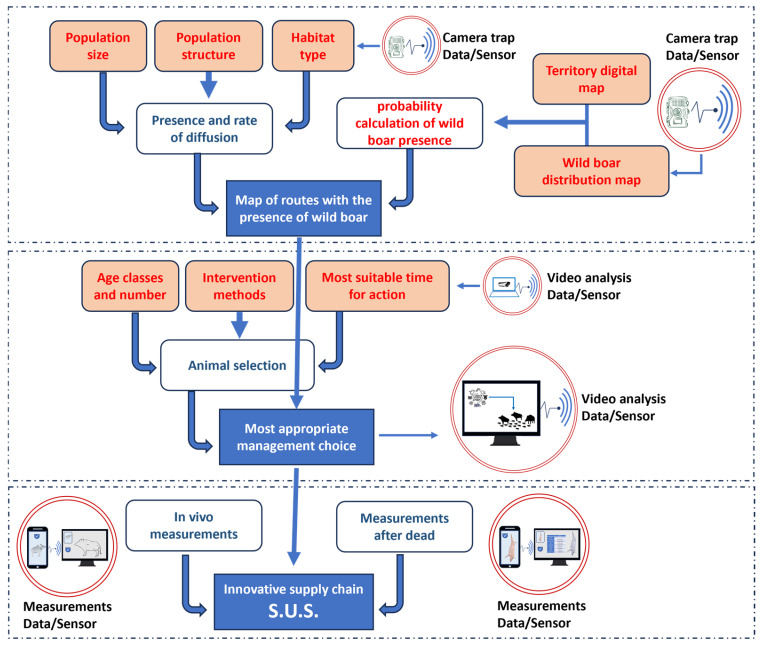
Complete architecture of the experimental structure, articulated in its three main phases (S.U.S. Campania).

**Figure 2 sensors-25-02071-f002:**
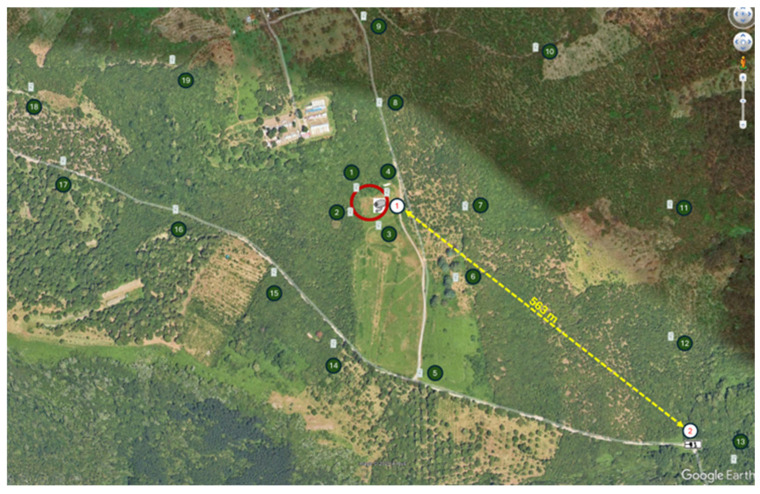
Distribution map of camera traps (1–19 black numbers), wireless Wi-Fi bridge (1–2 red numbers), selective capture cage placement area (red circle).

**Figure 3 sensors-25-02071-f003:**
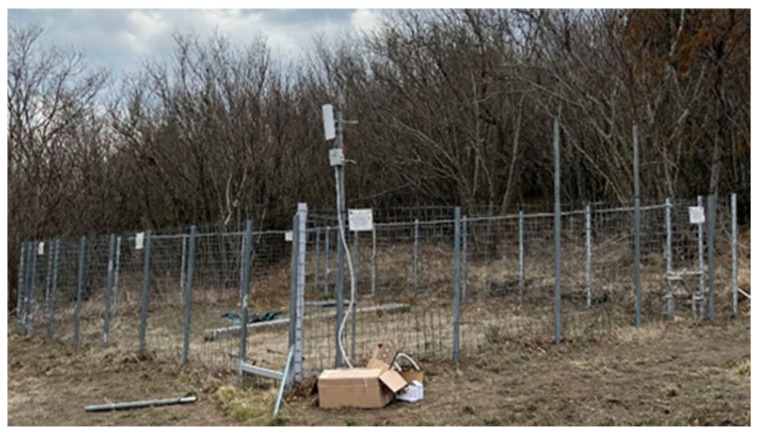
Innovative trap cage for the selective capture of sub-adult wild boars.

**Figure 4 sensors-25-02071-f004:**
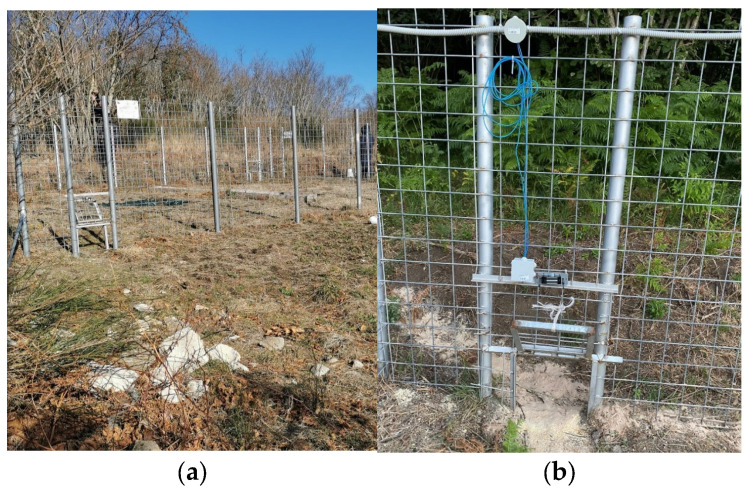
Capture cage structure: (**a**) in-field view of the trap cage; (**b**) detail of the selective overhead door.

**Figure 5 sensors-25-02071-f005:**
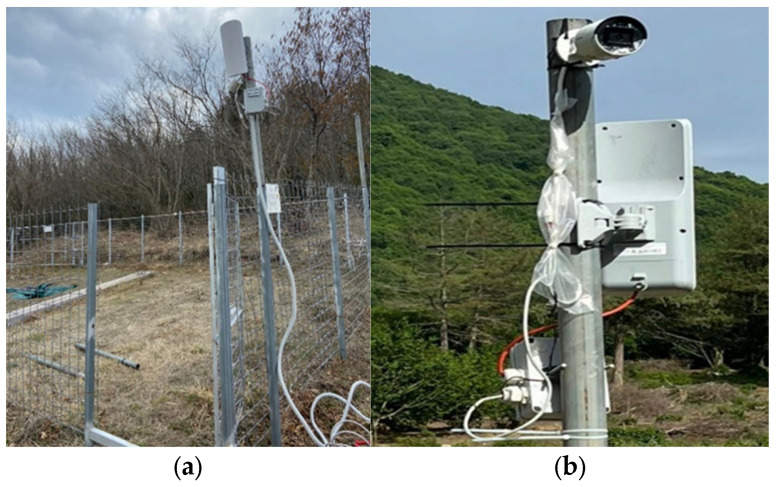
IoT video system: (**a**) integration of infrared camera with Wi-Fi Antenna of Wi-Fi bridge; (**b**) particular of infrared camera system connected with selective trap cage.

**Figure 6 sensors-25-02071-f006:**
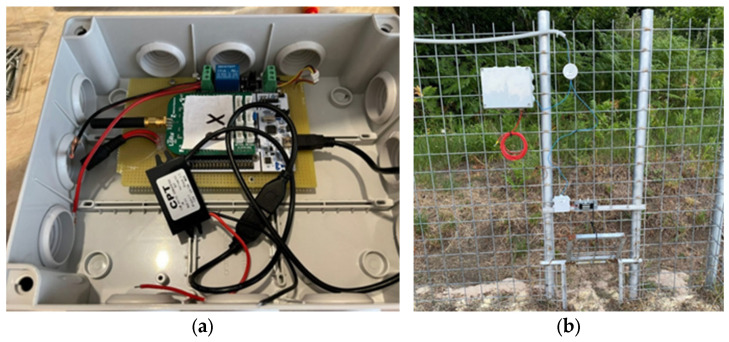
Overhead door mechanism: (**a**) remote control of the overhead door mechanism; (**b**) active overhead door for selective capture.

**Figure 7 sensors-25-02071-f007:**
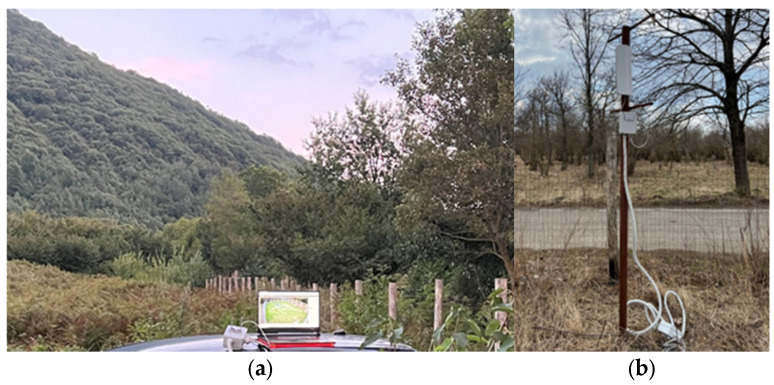
Wireless communication bridge system: (**a**) operators’ position 563 m from the capture cage; (**b**) particular of wireless communication bridge system.

**Figure 8 sensors-25-02071-f008:**
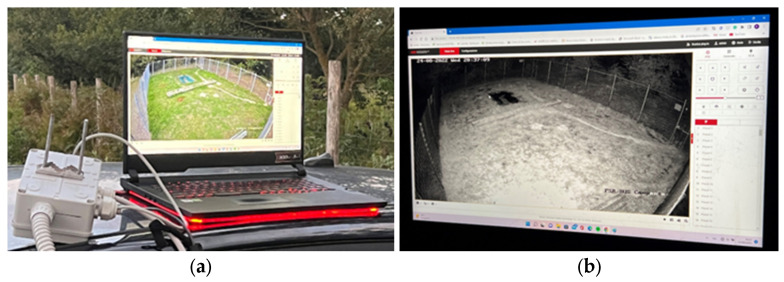
Remote control of capture cage activity: (**a**) daytime vision smart infrared camera system; (**b**) nocturne vision with a smart infrared camera system.

**Figure 9 sensors-25-02071-f009:**
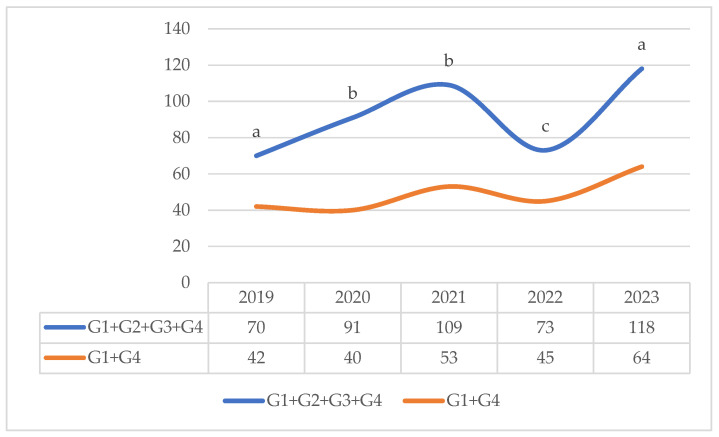
Trend of the wild boar population in the study area divided by age class (G1 = Young; G2 = Sub-adults; G3 = Young Adults; G4 = Adults) with or without the age classes G2 and G3. Lowercase (*p* ≤ 0.05) letters indicate significant differences among years.

**Table 1 sensors-25-02071-t001:** Number of wild boars identified in different age groups (G1 = Young; G2 = Sub-adults; G3 = Young Adults; G4 = Adults) with different video capture methods: Camera trap (CT) and IoT system (IoT).

Group Age (Months)	G1 1–6	G2 7–12	G3 12–24	G4 >24	Total
CT n.	244	175	105	58	582
CT %	42 ^Aa^	30 ^b^	18 ^cB^	10 ^B^	
IoT n.	198	138	79	46	461
IoT %	43 ^Aa^	30 ^b^	17 ^cB^	10 ^B^	

Uppercase (*p* ≤ 0.01) and lowercase (*p* ≤ 0.05) letters indicate significant differences among lines.

**Table 2 sensors-25-02071-t002:** Number of wild boars identified in different age groups and in different years of observation using the IoT system.

Year	2019	2020	2021	2022	2023	Total
G1	34	29	48	36	51	198
G2	15	36	41	11	35	138
G3	13	15	15	17	19	79
G4	8	11	5	9	13	46
Total	70	91	109	73	118	461

## Data Availability

The data that support the findings of this study are available from the corresponding author upon reasonable request.

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
