# Peer review of "Wild Boars’ Selective Capture with IoT and Electronic Devices Solutions for Innovative, Sustainable and Ethical Management"

_sensors, 2025, doi:10.3390/s25072071_

Round 1

Reviewer 1 Report

Comments and Suggestions for Authors

In this paper, the authors present a smart selective cage trap consisting of AI-based infrared cameras, LoRa communication and autonomous feeding systems. The device was used selective capture of wild boars in a study conducted over five years (2019 - 2023) in a Natura 2000 area in Campania, Italy.
The article mainly describes the capture device and presents the results of the field experiment in the form of two very succinct tables.

This article report an interesting piece of research, but the analysis made of it is far from satisfactory. Substantial improvements are needed.

Main comments:

1 - What do we know about the region's population of wild boards from other capture methods?

2 - There is a lack of a reference to compare the results of this approach

3 - The description of the results of the experiment is too brief. For example, how did the catches vary over time in the year? Seasonal effect? ​​Weather? Was there hunting in the area?

4 - What is the sensitivity and specificity of these smart traps? Costs?

5 - No assessment on stress or welfare of caught animal is done

Author Response

Ref.: Manuscript Number: Sensor-3469066

“Wild boars’ selective capture with IoT and electronic devices solutions for innovative, sustainable and ethical management” Electronic Sensors

Reviewer 1 comments:

In this paper, the authors present a smart selective cage trap consisting of AI-based infrared cameras, LoRa communication and autonomous feeding systems. The device was used selective capture of wild boars in a study conducted over five years (2019 - 2023) in a Natura 2000 area in Campania, Italy. The article mainly describes the capture device and presents the results of the field experiment in the form of two very succinct tables.

The authors thank referee1 for appreciating the experimental effort required when researchers working on a wildlife sample and for recognizing the results as worthy of publication in Sensors. The efforts made to remedy the defects highlighted by the referee to make the manuscript suitable for consideration for possible acceptance for publication are described and explained below.

This article report an interesting piece of research, but the analysis made of it is far from satisfactory. Substantial improvements are needed.

The authors thank Referee 1 for understanding the difficulties of scientific interventions related to wild animals. All comments produced by the review analysis will be taken into account to improve this research.

1 - What do we know about the region's population of wild boards from other capture methods?

At the moment, there are no published results regarding the numerical estimate of the wild boar population with capture methods.

The experimental area was chosen to demonstrate that a reliable numerical estimate of the wild boar population in a defined area was possible if the sum of the results obtained from different census methodologies were used. The Campania Region has invested in the Rural Development Plan 2014-2020 on the S.U.S. Campania project, the results of which were published on the page https://agricoltura.regione.campania.it/psr_2014_2020/1611_2/SUS.html.

The project was included in the PhD thesis in Veterinary Sciences by Dr. Piscopo Nadia entitled "Analysis and valuation of ecosystem services and their role for the conservation of biodiversity and socio-economic benefits in the agro-ecosystems of the Mediterranean Biogeographical Region".

The results estimated the number of wild boars in the years 2019-2022 which, on average, were:

- 7,018-8,214 wild boars in a large area of 31,800 hectares (STR18 - Monte Partenio - Monti di Avella - Pizzo d'Alvano.

- 130-247 wild boars in the focal area of 4.70 hectares (Antonio Raffaele farm).

2 - There is a lack of a reference to compare the results of this approach

There are currently no references to compare the approach proposed by the authors. This is confirmed by the fact that the proposed method is the subject of a patent.

3 - The description of the results of the experiment is too brief. For example, how did the catches vary over time in the year? Seasonal effect? Weather? Was there hunting in the area?

The authors emphasized the IoT system over the catches made since the result of greatest interest is the selective capture method rather than the number of animals captured. However, as recommended by the referee, more attention is paid to the results that are described in more detail regarding the variation in the catches over the years; the possible seasonal effect; the hunting period in different years.

4 - What is the sensitivity and specificity of these smart traps? Costs?

Thanks to the GitHub algorithm - Ultralytics YOLO8 system trained to recognize the morphological characteristics of wild boars and the experience of the authors, the sensitivity and specificity of the examined data was 74.57% (302/405 individual assessments).

The selective capture cage model was provided for by the PSR 2014-2020 funding and, being an experimental prototype to be patented, it cost Euro 20,000.

5 - No assessment on stress or welfare of caught animal is done

This aspect is the subject of a specific discussion that will be addressed in another publication.

Ref.: Manuscript Number: Sensor-3469066

“Wild boars’ selective capture with IoT and electronic devices solutions for innovative, sustainable and ethical management” Electronic Sensors

Reviewer 2 Report

Comments and Suggestions for Authors

This manuscript developed a selective capture system for wild boars using AI-based IoT and electronic devices. This is an interesting topic. As mentioned in this manuscript, the main goal was to collect real-time data on the behavior and movements of wild boars to provide precise indications before, during, and after the capture of animals. However, the following description in this version does not support achieving this goal technically sound. This manuscript descriptively introduced the features of each component of the hardware system in great detail. It lacks the details of the technical method or algorithm to support achieving the goals and needs scientific evaluation for the application performance of the developed system. Therefore, I would like to ask the authors for additions, corrections, and explanations.

Line 70: I wondered why it focused on capturing the population of subadults in 6-12 months rather than the population in the other age. Would you please give an explanation about this?

Line 96-105: How did the system process the thermal images to identify the animals presented in the images as the targeted age class and not as animals of other species? Please provide the algorithm framework. Does the system collect visual images?

Figure 1 & Line 81-92: How big is the experimental territory used in this study? How many camera traps and capture systems in total are used in the experimental area? This information is essential to assess the reliability of the obtained population size, territory digital map, etc. Suggest to provide a figure of the top view about the experimental area and the distribution of the camera traps, IoT Video System, and Capture cage, etc.

Line 157: What is the effective distance of LoRa communication and remote control used in the study?

Line 165-166: In Wireless Wi-Fi bridge, what is the distance of the wireless gateways? How many gateways used in total in this study?

Line 199: Please explain a little bit more about how the system assesses the meat quality using SWOT analysis.

Section 2.1: Suggest providing the description of the composition of the whole system used in this study. suggest to put figure 1 in section 2.1 and give a description of the goals/functions of each phase.

Section 2.2: Suggest giving more details on the method/algorithm to realize the goals of each phase. For example, the framework of image analysis to obtain the map of the route with the presence of wild boar, how did the authors judge the presence of target animals, the frame of SWOT analysis in meat quality assessment, etc.?

Line 216 & figure 2: suggest providing a schematic layout figure with dimensions of the trap cage and giving the introduction of the composition of the selective trap cage system.

Line 251-266: What algorithm did the author use to count the number of wild boars captured by the camera and estimate the population size? Any reference?

Section 2.3: Some components, such as a wireless Wi-Fi bridge, are introduced in both sections 2.2 and 2.3. suggest to merge with section 2.2 and then use subheadings to introduce each phase/system separately.

Section 2.4: Suggest merging with section 2.1.

Table 1: How did the camera and IoT system identify the age group of the person in the image or captured by the system? Have the camera traps and the IoT capture system been evaluated before, and how accurate are they?  Furthermore, how did the authors tell the age of the wild boar? Are there any distinct features of the boar that help to distinguish their age?

Table 2: How to evaluate the effect of using this boar capture system on the control of their population? From the data point of view, with the exception of 2022, the population of observed boar has been on the rise in the five-year period.

Author Response

Reviewer 2 comments:

This manuscript developed a selective capture system for wild boars using AI-based IoT and electronic devices. This is an interesting topic. As mentioned in this manuscript, the main goal was to collect real-time data on the behavior and movements of wild boars to provide precise indications before, during, and after the capture of animals. However, the following description in this version does not support achieving this goal technically sound. This manuscript descriptively introduced the features of each component of the hardware system in great detail. It lacks the details of the technical method or algorithm to support achieving the goals and needs scientific evaluation for the application performance of the developed system. Therefore, I would like to ask the authors for additions, corrections, and explanations.

The authors thank the referee2 for appreciating the experimental effort required when researchers work on a wildlife sample and for recognizing, after proper review, the results as worthy of publication in Sensors. The efforts made to remedy the shortcomings highlighted by the referee to make the manuscript eligible for consideration for possible acceptance for publication are described and explained below.

Line 70: I wondered why it focused on capturing the population of subadults in 6-12 months rather than the population in the other age. Would you please give an explanation about this?

The sentence: “To propose an innovative process for the management of wild boars’ population focused on subadults (6-12 months old), numerous surveys were conducted in the sample area through camera trapping [28].

has been modified also following

“The experimental scheme proposes an innovative process for the management of the wild boar population. All the methods proposed to date are based on the indiscriminate numerical reduction (no distinction between age classes and sex) carried out with firearms or by capturing the greatest possible number of subjects.

Our work proposes a demographic reduction of the population by operating only on the young age classes, so as not to destructure the family groups. In previous works [28], the presence of sub-adult wild boars (6-12 months of age) in the sample area was described by camera trapping [28].”

Line 96-105: How did the system process the thermal images to identify the animals presented in the images as the targeted age class and not as animals of other species? Please provide the algorithm framework. Does the system collect visual images?

The sentence: “A smart infrared thermal camera capable of detecting temperature variations and monitoring defined perimeters, with real-time analysis of anomalies and movements: this system, installed on one of the poles on the perimeter of the capture cage, allowed the detection of the type of animals standing inside the cage, as well as their counting and the definition of their relative age classes.

has been modified also following

An intelligent infrared thermal camera capable of detecting temperature variations and monitoring defined perimeters, with real-time analysis of anomalies and movements: this system, installed on one of the poles on the perimeter of the capture cage, allowed to detect the type of animals present inside the cage.

The counting of wild boars and the definition of their age classes was carried out using the GitHub algorithm - Ultralytics YOLO8 trained to recognize the morphological characteristics of wild boars (detection, object tracking, image classification).

Figure 1 & Line 81-92: How big is the experimental territory used in this study? How many camera traps and capture systems in total are used in the experimental area? This information is essential to assess the reliability of the obtained population size, territory digital map, etc. Suggest to provide a figure of the top view about the experimental area and the distribution of the camera traps, IoT Video System, and Capture cage, etc.

The details of what Referee 2 requested are reported in other works published by the authors. However, to respond to the review, the text "The use of camera traps made it possible to produce digital maps related to the distribution of suids in the examined agroecosystem."

has been modified as follows:

“The use of 19 camera traps made it possible to produce digital maps (Figure 2) related to the distribution of wild boars in the point area (4.70 hectares) and in the surrounding agroecosystem (61 hectares)”.

Figure 2 has been inserted with the following wording “Figure 2. Distribution map of camera traps in the study area”

Line 157: What is the effective distance of LoRa communication and remote control used in the study?

In this study, the LoRa communication and remote control had an effective transmission distance of 536 meters, considering the specific environmental conditions and the presence of obstacles. The system is designed for long-range communication, and nominally, in ideal line-of-sight conditions, LoRa can cover distances of up to 15 km

The sentence:

  • “Using a Wireless Wi-Fi bridge, the infrared camera transmits real-time images of the inside of the cage to a station located up to 5 km away. This setup is particularly beneficial in rural areas without telephone or Internet coverage, enabling constant monitoring and allowing specialized operators to intervene promptly when necessary [22].The decision to keep the image transmission channel and the remote door control channel separate, using Wi-Fi for real-time video streaming and LoRa for door closure commands, was made to enhance the system's reliability and prevent unwanted closures. By employing a dedicated wireless Wi-Fi bridge for transmitting infrared camera images, operators can continuously monitor the cage interior up to 5 km away without interfering with the LoRa-based control channel.

LoRa's low-bandwidth, long-range capabilities make it ideal for securely transmitting encrypted door control commands, while Wi-Fi's higher bandwidth is better suited for transmitting real-time video. This separation ensures that video transmission does not introduce delays or disruptions in the critical control commands for the door mechanism. Moreover, the two-step confirmation process for door closure commands, combined with end-to-end encryption, minimizes the risk of accidental or unauthorized activation.

This dual-channel architecture not only improves system efficiency and security but also ensures precise coordination, critical in scenarios where an untimely closure could jeopardize the entire operation.

has been modified also following

Using a Wireless Wi-Fi bridge, the infrared camera transmits real-time images of the inside of the cage to a remote station, making it an ideal solution for monitoring in areas without telephone or Internet coverage and allowing specialized operators to intervene promptly when necessary [22]. In this specific case, the effective distance was 563 meters in a straight line, with obstacles such as tall trees that could attenuate the signal (Figure 2). To ensure the connection, only two directional antennas were used, one for transmission and one for reception, without any intermediate repeaters. Figure 2 also indicates the lo-cations of the devices: Point 1 marks the position of the trap, while Point 2 corresponds to the remote receiving station. Despite environmental challenges, the system maintained a stable and effective transmission

Line 165-166: In Wireless Wi-Fi bridge, what is the distance of the wireless gateways? How many gateways used in total in this study?

In this study, the wireless gateways used in the Wi-Fi bridge setup had an effective transmission distance of 563 meters in a straight line, with obstacles such as tall trees. The system relied on a total of two gateways, one acting as the transmitting unit and the other as the receiving unit, both equipped with directional antennas to enhance signal strength and stability. No intermediate repeaters were used, yet the setup ensured reliable communication despite environmental challenges. Nominally, in this configuration, the system can cover distances of up to 5 km in line-of-sight conditions. This information has been incorporated into the text of the article, and a new figure has been added to illustrate the distance between the transmitter and the receiver.

Regarding lines 165-166 the text has been modified as reported above.

Line 199: Please explain a little bit more about how the system assesses the meat quality using SWOT analysis.

The details of what was requested by Referee 2 are reported in:

Esposito, L.; Viola, P.; Di Paolo, M.; Merino Goyenechea, L.J.; Marrone, R.; Altieri, D.; Primi, R.; Piscopo, N. The wild boar as an ecosystem service: moving steps towards biodiversity engineering. In Proceedings of 2nd Edition IEEE International Conference on Metrology for eXtended Reality, Artificial Intelligence and Neural Engineering, MetroXRAINE 2023, 893 – 898. Milano, Italy, (25 27 October 2023) 10.1109/MetroXRAINE58569.2023.10405739

Section 2.1: Suggest providing the description of the composition of the whole system used in this study. suggest to put figure 1 in section 2.1 and give a description of the goals/functions of each phase.

Starting with the suggestion of Referee 2, the authors decided to merge paragraphs 2.1 Experimental design and 2.2 Project phases into a single paragraph 2.1 Experimental design and project phases.

In this way, it is possible to describe step by step the objectives and functions of each phase of the experimentation.

Section 2.2: Suggest giving more details on the method/algorithm to realize the goals of each phase. For example, the framework of image analysis to obtain the map of the route with the presence of wild boar, how did the authors judge the presence of target animals, the frame of SWOT analysis in meat quality assessment, etc.?

Figure 1 schematically summarizes the three phases on which the authors' work is structured and a brief description is provided in paragraph 2.1. It did not seem appropriate to increase the text with data and results relating to phase 1 and phase 3, already described by the authors in other publications [28, 29]. The objectives and the image analysis framework to obtain the route map with the presence of wild boars were achieved thanks to the help of the GitHub algorithm - Ultralytics YOLO8 technology which allowed us to identify the morphological characteristics and therefore the age classes of the photo-trapped wild boars.

The population structure was obtained thanks to the authors' skills. The details are reported in paragraph 2.1

The SWOT work framework is reported in the work [29]. Since this work deals with phase 2 and not phase 3 of the SUS project, the authors do not consider it appropriate to dwell on topics published elsewhere.

The sentence “The action scheme is summarized in the next sections” has been deleted.

Line 216 & figure 2: suggest providing a schematic layout figure with dimensions of the trap cage and giving the introduction of the composition of the selective trap cage system.

The trap cage schematic is pending patent and therefore cannot be published in the text. However, it is provided with supplementary material.

Line 251-266: What algorithm did the author use to count the number of wild boars captured by the camera and estimate the population size? Any reference?

The data for this observation was provided in paragraph 2.1. Experimental design and project phases

Section 2.3: Some components, such as a wireless Wi-Fi bridge, are introduced in both sections 2.2 and 2.3. suggest to merge with section 2.2 and then use subheadings to introduce each phase/system separately.

The issue has been resolved by merging sections 2.2 and 2.3 into a single section. Subheadings have been introduced to clearly distinguish each phase/system while maintaining a logical flow. This ensures a more cohesive structure and avoids redundancy, improving the overall readability of the document.

Section 2.4: Suggest merging with section 2.1

As suggested by Referee2, section 2.4 has been incorporated into section 2.1

Table 1: How did the camera and IoT system identify the age group of the person in the image or captured by the system? Have the camera traps and the IoT capture system been evaluated before, and how accurate are they?  Furthermore, how did the authors tell the age of the wild boar? Are there any distinct features of the boar that help to distinguish their age?

The identification of the age group was obtained by the analysis of the images using the GitHub algorithm - Ultralytics YOLO8 which recognize the morphological characteristics of wild boars and, properly trained, ensures accurate and reliable results. The distinctive characteristics of wild boars were defined thanks to the bibliographic indications [36, 38] and the consolidated experience of the authors [28].

Table 2: How to evaluate the effect of using this boar capture system on the control of their population? From the data point of view, with the exception of 2022, the population of observed boar has been on the rise in the five-year period.

The observation of Referee 2 is very pertinent. Let's try to explain:

The wild boars that entered the capture cage are those of the G2 and G3 groups which, if subtracted from the total number of subjects counted for the different age classes, would reduce the population to the sum of the adult and very young wild boars. Graph 1 demonstrates what has been stated and the differences are all statistically significant.

The authors considered inserting the following sentence:

Graph 1 shows that if the subadults (G2 and G3), the only ones that can enter the selective capture cage, are removed from the existing wild boar population, the remaining population is contained 40%, 56%, 51%, 38%, 46% respectively in the years 2019, 2020, 2021, 2022 and 2023 (P<0.05).

Round 2

Reviewer 1 Report

Comments and Suggestions for Authors

The following author's reponses should be included in the manuscript:

4 - What is the sensitivity and specificity of these smart traps? Costs?
Thanks to the GitHub algorithm - Ultralytics YOLO8 system trained to recognize the morphological characteristics of wild boars and the experience of the authors, the sensitivity and specificity of the examined data was 74.57% (302/405 individual assessments).
The selective capture cage model was provided for by the PSR 2014-2020 funding and, being an experimental prototype to be patented, it cost Euro 20,000.
5 - No assessment on stress or welfare of caught animal is done
This aspect is the subject of a specific discussion that will be addressed in another publication.

However, concerning the question of specificity and sensitivity of the authors have not satisfactorily addressed that. They should read what that means and to address it and then redo the calculations and report it in the manuscript.

In addition, revise the caption of Graph1 as follows: add the meaning G1, ..., G4 and provide the meaning of annotated letters: a, b and c

Author Response

Ref.: Manuscript Number: Sensor-3469066

“Wild boars’ selective capture with IoT and electronic devices solutions for innovative, sustainable and ethical management” Electronic Sensors

Reviewer 1 comments round 2:

The authors thank referee 1 for accepting the responses to round 1 and respond to the comments made in round 2.

The following author's reponses should be included in the manuscript:

 4 - What is the sensitivity and specificity of these smart traps? Costs?

Thanks to the GitHub algorithm - Ultralytics YOLO8 system trained to recognize the morphological characteristics of wild boars and the experience of the authors, the sensitivity and specificity of the examined data was 74.57% (302/405 individual assessments).

The selective capture cage model was provided for by the PSR 2014-2020 funding and, being an experimental prototype to be patented, it cost Euro 20,000.

5 - No assessment on stress or welfare of caught animal is done

This aspect is the subject of a specific discussion that will be addressed in another publication.

 However, concerning the question of specificity and sensitivity of the authors have not satisfactorily addressed that. They should read what that means and to address it and then redo the calculations and report it in the manuscript.

 Regarding the question of specificity and sensitivity, the authors report the calculation table for sensitivity and specificity:

Camera trap performance

Subadults (detected as)

Subadults (not detected as)

Wild boars

Subadult wild boars

a (True Positive)

c (False Negative)

No subadult wild boars

b (False Positive)

d (True Negative)

a = true-positive: Subadult wild boars entered in the cage and recognized by the camera trap (G2 + G3);

b = false-positive: No subadult wild boars entered in the cage and recognized by the camera trap (G1);

c = false-negative: Subadult wild boars entered in the cage and not recognized by the camera trap (G2 + G3);

d = true-negative: No subadult wild boars entered in the cage and not recognized by the camera trap (G1).

Camera trap performance

Subadults (detected as)

Subadults (not detected as)

Wild boars

Subadult wild boars

217

74

No subadult wild boars

198

148

Sensitivity (recall): a/(a+c)

217/(217+74)

0.745704467

Specificity: d/(b+d)

148/(198+148)

0.572254335

Alfa error: b/(b+d)

198/(198+148)

0.572254335

Beta error: c/(a+c)

74/(217+74)

0.745704467

Positive predictive value (precision): a/(a+b)

217/(217+198)

0.522891566

Negative predictive value: d/(c+d)

148/(74+148)

0.333333333

 In the manuscript, the following sentence was added to the results:

The system trained to recognize the morphological characteristics of wild boars and the experience of the authors showed the sensitivity was 74.57% while the specificity was lower (57.22%).

  In addition, revise the caption of Graph1 as follows: add the meaning G1, ..., G4 and provide the meaning of annotated letters: a, b and c

The authors have revised the caption of Graph 1 and added the meanings of the respective acronyms.

Reviewer 2 Report

Comments and Suggestions for Authors

The authors responded and justified all points of the review. Please make sure to proofread the manuscript again and check the consecutiveness of the figure number and section number. If possible, I still suggest the authors give the accuracy of the system on recognizing the different age groups of wild boars in the results section.

Author Response

Ref.: Manuscript Number: Sensor-3469066

“Wild boars’ selective capture with IoT and electronic devices solutions for innovative, sustainable and ethical management” Electronic Sensors

Reviewer 2 comments round 2:

The authors thank referee 2 for accepting the responses to round 1 and respond to the comments made in round 2.

The authors have responded and justified all points of the review. Be sure to reread the manuscript and check the consecutiveness of the figure number and section number. If possible, I still suggest that the authors indicate the accuracy of the system in recognizing the different age groups of wild boars in the results section.

The authors proofread the manuscript and checked the consecutiveness of the figure number and section number. The authors indicate the accuracy of the system in point 2.2.

In the manuscript, the following sentence was added to the results:

The system trained to recognize the morphological characteristics of wild boars and the experience of the authors showed the sensitivity was 74.57% while the specificity was lower (57.22%).
